# SkillBERT: "Skilling" the BERT to classify skills using Electronic Recruitment Records

**Amber Nigam**
Harvard University
Cambridge, MA, USA

**Shikha Tyagi**
PeopleStrong
New Delhi, India

**Kuldeep Dhakar**
Peoplestrong
New Delhi, India

## Abstract

In this work, we show how the Electronic Recruitment Records (ERRs) that store the information related to job postings and candidates can be mined and analyzed to provide assistance to hiring managers in recruitment. These ERRs are captured through our recruitment portal, where hiring managers can post the jobs and candidates can apply for various job postings. These ERRs are stored in the form of tables in our recruitment database and whenever there is a new job posting, a new ERR is added to the database.

We have leveraged the skills present in the ERRs to train a BERT-based model, SkillBERT, the embeddings of which are used as features for classifying skills into groups referred to as "competency groups". A competency group is a group of similar skills, and it is used as matching criteria (instead of matching on skills) for finding the overlap of skills between the candidates and the jobs. This proxy match takes advantage of the BERT's capability of deriving meaning from the structure of competency groups present in the skill dataset. The skill classification is a multi-label classification problem as a single skill can belong to multiple competency groups. To solve multi-label competency group classification using a binary classifier, we have paired each skill with each competency group and tried to predict the probability of that skill belonging to that particular competency group. SkillBERT, which is trained from scratch on the skills present in job requisitions, is shown to be better performing than the pre-trained BERT (Devlin et al., 2019) and the Word2Vec (Mikolov et al., 2013). We have also explored K-means clustering (Lloyd, 1982) and spectral clustering (Chung, 1997) on SkillBERT embeddings to generate cluster-based features. Both algorithms provide similar performance benefits. Last, we have experimented with different classification models like Random Forest (Breiman, 2001), XGBoost (Chen and Guestrin, 2016), and a deep learning algorithm Bi-LSTM (Schuster and Paliwal, 1997; Hochreiter and Schmidhuber, 1997) for the tagging of competency groups to skill. We did not observe a significant performance difference among the algorithms, although XGBoost and Bi-LSTM perform slightly better than Random Forest. The features created using SkillBERT are most predictive in the classification task, which demonstrates that the SkillBERT is able to capture the information pertaining to skill ontology from the data. We have made the source code, the trained models, and the dataset [1] of our experiments publicly available.

---

[1] https://www.dropbox.com/s/wcg8kbq5btl4gm0/code_data_pickle_files.zip

Submitted to the 35th Conference on Neural Information Processing Systems (NeurIPS 2021) Track on Datasets and Benchmarks. Do not distribute.

# 1 Introduction

A Competency group can be thought of as a group of similar skills required for success in a job. For example, skills such as *Apache Hadoop*, *Apache Pig* represent competency in Big Data analysis while *HTML*, *Javascript* are part of Front-end competency. Classification of skills into the right competency groups can help in gauging a candidate's job interest and automation of the recruitment process.

Recently, there has been a surge in online recruitment activity. The researchers are using the data available through these online channels to find patterns in the skills of candidates and jobs. Several semantic approaches are also being used to minimise the manual labour required in the recruitment industry.

Bian et al. (2019) proposed a system to match the sentences from job posting and candidate resume using a deep global match network. They proposed a system which consists of finding the sentence-level representation. The representation is then used for the sentence-level match and global match. The experiments conducted on a large corpus showed the effectiveness of the model, especially in the cases of labeled data scarcity.

Ozcaglar et al. (2019) proposed an entity-personalized Talent Search model which utilizes a combination of generalized linear mixed (GLMix) models and gradient boosted decision tree (GBDT) models, and provides personalized talent recommendations using nonlinear tree interaction features generated by the GBDT. They have also presented an architecture for online and offline productionization of this hybrid model.

Qin et al. (2018) developed an Ability-aware Person-Job Fit Neural Network (APJFNN) model to minimize the dependence on manual work in the recruitment industry. They used a large corpus of historical job application data and developed a Recurrent Neural Network (RNN) based model on job requirements and job seekers' experiences to learn a word-level semantic representation. They implemented four hierarchical ability-aware attention strategies with an aim to learn the importance of job requirements based on the semantics. They also measured the job experience contribution for specific ability requirements.

Xu et al. (2018) in their work measured the popularity of the job skills in the recruitment market using a multi-criteria approach. They explored a huge corpus of job postings and constructed a job skill network. Using this network, they developed a novel Skill Popularity based Topic Model (SPTM). By using SPTM, they were able to use multiple criteria of jobs such as salary level and company size, and latent connections within skills. They utilized the multi-faceted popularity of the job skills for rank ordering.

Alabdulkareem et al. (2018) have used skill topology and connection between skills to explain dynamics such as the transition between occupations by workers, the comparative advantage of certain cities in new skills, and change in skill requirement as per occupation. By using unsupervised clustering techniques, they have shown that two clusters are formed where one represents the social-cognitive skills and the second represents sensory-physical skills that belong to high and low-wage occupations, respectively.

For learning features from the text data, several contextual word embedding models have been explored on various domain-specific datasets but no work has been done on exploring those models on job-skill specific datasets.

Fields like medical and law have already explored these models in their respective domains. Lee et al. (2019) in their BioBERT model trained the BERT model on a large biomedical corpus. They found that without changing the architecture too much across tasks, BioBERT beats BERT and previous state-of-the-art models in several biomedical text mining tasks by a large difference. Alsentzer et al. (2019) trained publicly released BERT-Base and BioBERT-finetuned models on clinical notes and discharge summaries. They have shown that embeddings formed are superior to a general domain or BioBERT specific embeddings for two well-established clinical NER tasks and one medical natural language inference task (i2b2 2010 (Uzuner et al., 2011), i2b2 2012 (Sun et al., 2013a,b)), and MedNLI (Romanov and Shivade, 2018)).

Beltagy et al. (2019) in their model SciBERT leveraged unsupervised pretraining of a BERT based model on a large multi-domain corpus of scientific publications. SciBERT significantly outperformed

BERT-Base and achieves better results on tasks like sequence tagging, sentence classification, and dependency parsing, even compared to some reported BioBERT results on biomedical tasks.

Similarly, Elwany et al. (2019) in their work has shown the improvement in results on fine-tuning the BERT model on legal domain-specific corpora. They concluded that fine-tuning BERT gives the best performance and reduces the need for a more sophisticated architecture and/or features.

In this paper, we are proposing a competency group classifier, which primarily leverages: SkillBERT, which uses BERT architecture and is trained on the job-skill data from scratch to generate embeddings for skills. These embeddings are used to create several similarity-based features to capture the association between skills and group. We have also engineered features through clustering algorithms like spectral clustering on embeddings to attach cluster labels to skills. All these features along with SkillBERT embeddings are used in the final classifier to achieve the best possible classification accuracy.

## 2 Methodology

As no prior benchmark related to job-skill classification is available, we manually assigned each skill in our dataset to one or more competency groups with the help of the respective domain experts. We experimented with three different models: pre-trained BERT, Word2vec, and SkillBERT to generate skill embeddings. Word2vec and SkillBERT were trained from scratch on our skill dataset. We created similarity-based and cluster-based features on top of these embeddings. The details of dataset design and feature engineering used for model creation are given in the next sections.

### 2.1 Training data creation

As no prior competency group tagging was available for existing skills, we had to manually assign labels for training data creation. For this task, the skill dataset is taken from our organization's database which contains 700,000 job requisitions and 2,997 unique skills. The competency groups were created in consultation with domain experts across all major sectors. Currently, there exists 40 competency groups in our data representing all major industries. Also within a competency group, we have classified a skill as *core* or *fringe*. For example, in *marketing* competency group, *digital marketing* is a *core* skill while *creativity* is a *fringe* skill.

Following instructions were given to the domain experts for the annotation exercise:

1. A skill can belong to multiple groups

2. If they are unable to recognize a skill, they may annotate it based on the knowledge gathered from searching about it on the internet

3. If a skill belongs to a particular group, then the experts must further classify it as a core(strongly related) or fringe(weakly related) skill to that group

The mapping of competency groups and skills can be downloaded here. Table 1 contains examples of some candidate and job profiles.

Once training data is created, our job is to classify a new skill into these 40 competency groups. Some skills can belong to more than one category also. For such cases, a skill will have representation in multiple groups. Figure 1 shows an overview of the datasets used in this step.

### 2.2 Feature Engineering

For feature creation, we have experimented with Word2vec and BERT to generate skill embeddings. By leveraging these skill embeddings we created similarity-based features as well. We also used clustering on generated embeddings to create cluster-based features. As multiple clustering algorithms are available in the literature, we evaluated the most popular clustering algorithms – K-means (Lloyd, 1982) and spectral clustering for experimentation. We have done extensive feature engineering to capture information at skill level, group level, and skill-group combination level. The details of features designed for experiments are given below.

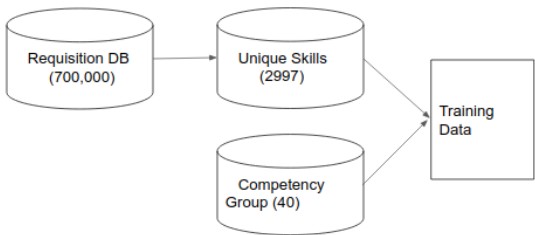

Figure 1: Dataset used for training data creation

Table 1: Examples of some candidate and job profiles

| Candidate or Job | Skill set | Competency groups |
|---|---|---|
| Candidate1 | Design, KnockoutJS, CorelDRAW | Tool design, Mechanical design, Front end, Web development |
| Candidate2 | Statistical modeling, Statistical process control | Statistics, Production operations |
| Job1 | Analytical skills, Project execution, Accounting | Financial operations, Business analytics, Statistics, Accounts |
| Job2 | Digital marketing, Cash management, MS Office, MS Excel, MS Word, Tally | Taxation, Banking, Statistics |

### 2.2.1 Embedding features:

Traditionally, n-gram based algorithms were used to extract information from text. However, these methods completely ignore the context surrounding a word. Hence, we have experimented with Word2vec and BERT based architecture to learn embeddings of skills present in training data. The details of how we have leveraged them in our problem domain are given below.

**Word2vec** uses a shallow, two-layer neural network to generate n-dimensional embedding for words. To use the Word2vec model on requisition data, we extracted skills from job requisitions and constructed a single document. Each sentence of this document represents the skills present in one requisition. As a requisition can have multiple skills, we created a 2-dimensional list, where the outer dimension specifies the sentence and the inner dimension corresponds to the skills in that sentence. E.g. if there are two job requisitions called req1 and req2 and their corresponding skills are "Java, J2EE" and "Logistic regression, Data visualization, NLP" then outer index 0 corresponds to req1 and outer index 1 corresponds to req2. Index 0,0 will refer to Java and Index 0,1 will refer to J2EE and so on. Also before feeding this data for training lowercasing of words, stop word removal and stemming was performed as part of preprocessing. A total of more than 700,000 requisitions were used for model training. We have used embeddings of size 30 which was decided after evaluating model performance on different embedding sizes.

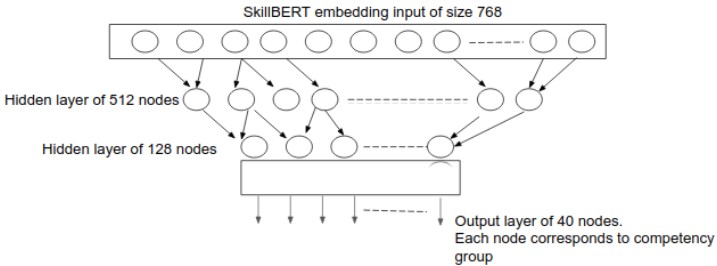

Figure 2: Classifier architecture

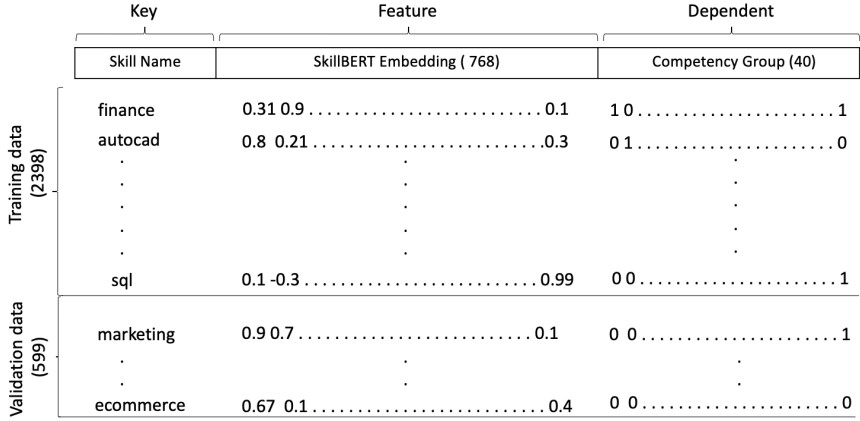

Figure 3: Data format used for creating bert_ prob feature

**BERT** Bidirectional Encoder Representations from Transformers, is designed to pre-train deep bidirectional representations from the unlabeled text by jointly conditioning on both left and right context in all layers. The pre-trained BERT model can be fine-tuned with just one additional output layer to create state-of-the-art models for tasks such as question answering, next sentence prediction, etc. Similar to Word2vec, BERT can also be used to extract fixed-length embeddings of words and sentences, which can further be used as features for downstream tasks like classification. But unlike fixed embedding produced by Word2vec, BERT will generate different embedding for an input word based on its left and right context. BERT has shown performance improvement for many natural language processing tasks. However, it has been minimally explored on the job-skill database. Hence, we leveraged BERT architecture on skill data to train the SkillBERT model. We have used AWS cloud machine type:*ml.p2.xlarge* with GPU memory *12 GiB* and processor *1xK80* GPU for SkillBERT training and it took us around 72 hours to completely train it on our dataset. In the next section, we have given the details of training BERT on skill corpus.

**Training:** For training BERT, we used the same corpus as used for Word2vec training and experimented with hyperparameters like learning rate and maximum sequence length. For the learning rate, we used 0.1, 0.05, and 0.01 and finalised 0.01. For maximum sequence length, we used 64,128, 180 and finalised 128. We could not perform extensive hyperparameter tuning due to hardware limitations. Once the training is finished, we extract the last hidden layer output of size 768 and further reduce the embedding size to 128 to decrease the training time of our final model discussed in the experiment section. For the dimensionality reduction of embedding, we did experiments with embeddings of sizes 32, 64, 128 and 256. As shown in Appendix Table 7, the best results were obtained using embedding of size 128. To make sure information from all the 768 dimensions is leveraged, we trained a 2-layer neural network classifier using SkillBERT embeddings as an independent feature and competency group as a dependent variable. Out of the 2,997 skills, 80% were used for training and the rest of the 20% were used for the validation. This model generates the probability values of a skill belonging to each of the 40 competency groups and was used as a feature in the final model at skill and competency group combination level. We have referred to this feature as "bert-prob" in the rest of sections. The architecture of the model used for getting these probabilities is shown in Figure 2 and Figure 3 represents the data format used to generate the bert- prob feature.

### 2.2.2 Similarity-based features:

By leveraging skill embeddings generated using embedding techniques, similarity-based features were created to capture the association between a group and skill. The details of those are given below.

**Similarity from competency group:** In competency group data, the name of each competency group is also present as a skill. We created a similarity score feature measuring the cosine distance between competency group name and skill embeddings.

Table 2: Machine Learning model Hyperparameters

| Machine Learning Models | Best Hyperparameters | Hyperperameter bound |
|---|---|---|
| XGBoost | N_estimators:800, Depth:5 | N_estimators:400-1000, Depth:3-7 |
| Random Forest | N_estimators:700, Depth:4 | N_estimators:400-1000, Depth:3-7 |
| Bi-LSTM | Layers:2(Nodes: 128, 64), Optimizer:Adam, Dropout:0.2 | Layers:2 - 4 , Nodes: 32 - 512, Dropout: 0.1 - 0.5 |

Table 3: Machine Learning model training time

| Machine Learning Models | Training Time (in seconds) |
|---|---|
| XGBoost | 122 |
| Random Forest | 87 |
| Bi-LSTM | 167 |

**Similarity from top skills per group:** Apart from utilizing the similarity between competency group name and skill, we have also created similarity-based features between a given skill and skills present in the competency group. As an example, we have a skill named *auditing* and competency group *finance*. Three similarity-based features were created called top1, top2, and top3, where top1 is cosine similarity score between skill *auditing* and most similar skill from *finance*, top2 is the average cosine similarity score of top two most similar skills and top3 is the average cosine similarity score of top three most similar skills. As shown in Appendix Table 6, the use of similarity-based features beyond three skills did not improve model performance.

### 2.2.3   Cluster-based feature:

For generating labels for skills using clustering, we experimented with two techniques on SkillBERT embedding – *K-means* and *spectral clustering*. Scikit-learn package of K-means was used to generate 45 cluster labels. The number 45 was decided by using the elbow method, the graph of which is shown in Appendix Figure 9. 35 cluster labels were generated using spectral clustering. The number 35 was decided on the basis of the "gap" in the smallest eigenvalues. The details of how we used spectral clustering on SkillBERT embedding to generate cluster-based feature are given in Appendix section A.1.

### 2.2.4   Skill TFIDF feature:

TFIDF (Salton and McGill, 1986; Ramos, 1999) is widely used in text mining to find rare and important words in a document, and as in our training data a single skill can be part of multiple competency groups, we used the same strategy to find skills that are unique to a competency group by calculating their TFIDF value. However, as group information will not be available for new skills, we will calculate the TFIDF of such skills differently. First, we will find the most similar top 3 existing skills and thereafter, take the average of their TFIDF values. This resultant value will be the TFIDF value for a new skill.

### 2.2.5   Core and fringe skills:

Apart from the features mentioned in the above sections, we have also created group-based features by counting the number of core and fringe skills in each group.

## 3   Experiments

Though the categorization of skills into multiple competency groups is a multi-label classification problem, we have approached this as a binary classification problem by preparing our training data as skill-competency group pairs i.e. for each skill we will have 40 rows, corresponding to each competency group. For each skill-competency group pair, we have tried to predict the probability of that skill belonging to that competency group using classifier models like XGBoost, Random Forest, and Bi-LSTM. Pairs of models which were compared and had a statistically significant difference in

| | Skill Name(2997) X competency Group(40) | SkillBERT Embedding (128) | bert_prob | spectral cluster index | TFIDF | bert_grp_sim | skill-skill similarity(3) | fringe_skill_count | core_skill_count | 0/1 |
|---|---|---|---|---|---|---|---|---|---|---|
| Training data(~96K) | ppc, digital marketing | 0.31 ..0.1 | 1 | 1 | 0.4 | 0.97 | 0.97,0.85,0.8 | 10 | 20 | 1 |
| | ppc, finance | 0.31 ..0.1 | 0 | 1 | 0.4 | 0.63 | 0.63,0.5,0.4 | 3 | 15 | 0 |
| | . | . | . | . | . | . | . | . | . | . |
| | . | . | . | . | . | . | . | . | . | . |
| | . | . | . | . | . | . | . | . | . | . |
| | pig, big data | 0.11 ..0.2 | 1 | 5 | 0.2 | 0.89 | 0.89,0.82,0.6 | 5 | 12 | 1 |
| Validation data (~24K) | html, web development | 0.91 ..0.01 | 0.9 | 4 | 0.2 | 0.75 | 0.91,0.75,0.8 | 3 | 14 | 1 |
| | . | . | . | . | . | . | . | . | . | . |
| | nlp, machine learning | 0.22 ..0.1 | 0.8 | 9 | 0.7 | 0.78 | 0.8,0.79,0.7 | 9 | 25 | 1 |

Figure 4: Data format used for final model creation

the performance are highlighted with a star in Table 4. The data format used for final model is shown in Figure 4 and the details of all the experiments are given below.

**SkillBERT vs Word2vec vs Pre-trained BERT:** As the first experiment, we did a comparative study among SkillBERT, pre-trained BERT, and Word2vec models. For pre-trained BERT, we used the "bert-base-uncased" model which also produces embeddings of size 768. Similar to SkillBERT, we reduced the embedding size to 128 and generated "bert-prob" feature. All features except cluster labels discussed in the feature engineering section were created using these embedding models. To better analyze the quality of embeddings, we projected high dimensional embeddings of skills present in competency groups in 2-D using t-SNE (van der Maaten and Hinton, 2008). From visualization shown in Figure 5 and Figure 6, it is clear that Skill-BERT embeddings reduced the overlapping gap between groups and gave well-defined cluster boundaries as compared to word2vec and pre-trained BERT. As a classifier, we used XGBoost and performed hyperparameter tuning through grid-search to get the best possible result without over-fitting. In the training dataset, there was a total of 95,904 records and 2,398 unique skills while the testing dataset had 23,976 records and 599 unique skills. The results of this experiment are given in Table 4.

**K-means vs spectral clustering:** In this experiment, we tried to see the effect of adding cluster-based features generated using K-means and spectral clustering on SkillBERT embedding. For this comparison, we applied XGBoost on the cluster labels and the features used in the previous experiment where we compared different embedding approaches.

**Random Forest vs Bi-LSTM vs XGBoost:** As part of this experiment, we applied Bi-LSTM, Random Forest, XGBoost, and spectral clustering based features on SkillBERT and compared their performance. Table 2 contains the best performing hyperparameter values and their variation range during tuning through grid-search for all the classifiers used. The number of hyperparameter search trials done was 20, 20, 36 for XGBOOST, Random Forest, and Bi-LSTM models respectively. Table 3 contains the training time of each classifier model.

**Core vs fringe skill classification:** Finally, we also trained a multi-class classifier to see how accurately we can classify *core* and *fringe* skills. For this, we trained a model with 3 classes where, class 0 – *no label*, class 1 – *fringe skill*, and class 2 – *core skill*. All the features used in the last experiment were leveraged for this experiment and Bi-LSTM was used as a classifier.

**Impact evaluation:** While screening the candidate resumes, hiring managers come across many skills that are unknown to them. For such skills, they invest time in searching the domain. By normalizing the skills to the competency groups with the help of SkillBERT, we are reducing the

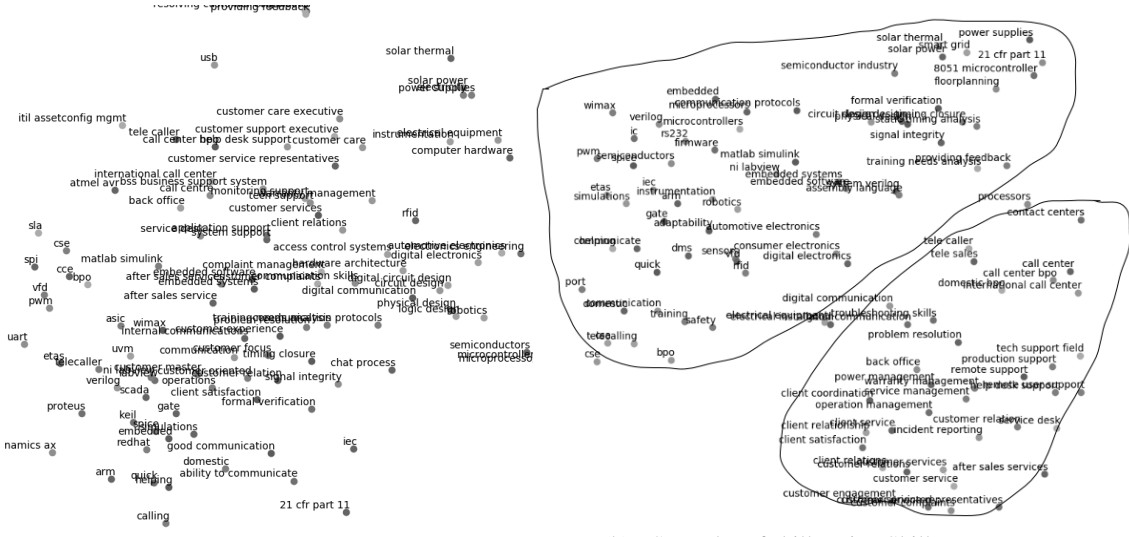

(a) t-SNE plot of skills using pre-trained BERT

(b) t-SNE plot of skills using SkillBERT

Figure 5: t-SNE plot of embeddings of "Customer Support" and "Electronics" competency group. The left image shows the projection generated using pre-trained BERT embedding and the right image is the SkillBERT plot. The top cluster shown in SkillBERT t-SNE plot represents "Electronics" competency group while the bottom cluster represents "Customer Support".

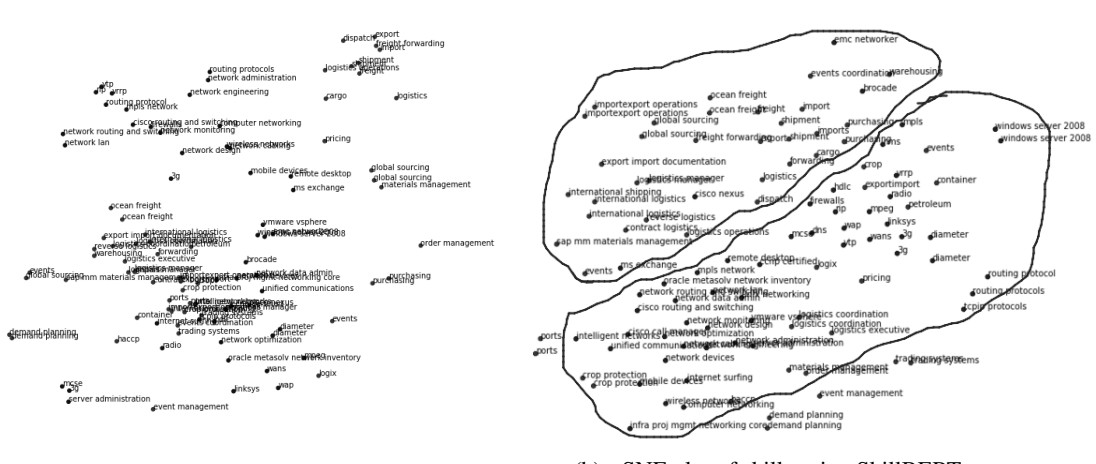

(a) t-SNE plot of skills using Word2vec

(b) t-SNE plot of skills using SkillBERT

Figure 6: t-SNE plot of embeddings of "Logistic" and "Network" competency group. The left image shows the projection generated using Word2vec embedding and the right image is the SkillBERT plot. The top cluster shown in SkillBERT t-SNE plot represents "Logistic" competency group while the bottom cluster represents "Network".

time taken by the hiring managers to find the domain of the skills and consequently reducing the screening time of resumes. The difference in time is because the SkillBERT not only matches the skills to their domains (groups), but it also shows constituent skills in each group, thereby providing more context about the groups to the hiring managers and thus reducing their search-time. As of now, there is no automated way of tracking the resume screening rate on our platform. However, it has been observed that there is a 150% increase in the number of average resumes screened per day after the introduction of SkillBERT. The above metric does not account for the confounders like hiring manager's experience and performance among other covariates.

Table 4: Evaluation of results on different embedding models and feature sets

| Model | Precision | | Recall | | F1-score | |
|---|---|---|---|---|---|---|
| | Class 0 | Class 1 | Class 0 | Class 1 | Class 0 | Class 1 |
| *XGBoost + pre-trained BERT | 98.83% | 51.54% | 95.85% | 74.26% | 97.21% | 60.84% |
| *XGBoost + Word2vec | 98.06% | 68.34% | 97.36% | 65.21% | 96.53% | 66.73% |
| XGBoost + SkillBERT | 99.32% | 96.65% | 99.47% | 84.82% | 99.39% | 90.35% |
| XGBoost + SkillBERT + K-means | 99.27% | 96.92% | 99.54% | 85.24% | 99.40% | 90.70% |
| Random Forest + SkillBERT + spectral clustering | 99.28% | 95.15 % | 99.50% | 83.48% | 99.39% | 88.93% |
| XGBoost + SkillBERT + spectral clustering | 99.35% | 97.23% | 99.48% | 85.09% | **99.41%** | **90.76%** |
| *Bi-LSTM + SkillBERT + spectral clustering | 99.26% | 95.86% | 99.57% | 86.43% | **99.42%** | **90.90%** |

Table 5: Core vs fringe skill classifier results

| Precision | | | Recall | | | F1-score | | |
|---|---|---|---|---|---|---|---|---|
| Class 0 | Class 1 | Class 2 | Class 0 | Class 1 | Class 2 | Class 0 | Class 1 | Class 2 |
| 99.07% | 93.19% | 99.76% | 99.74% | 78.28% | 62.45% | 99.40% | 85.08% | 76.81% |

## 4   Results

Results shown in Table 4 for competency group classification show that SkillBERT improved the performance of the classification model over Word2vec and pre-trained BERT. Use of XGBoost with SkillBERT based features give an F1-score of 90.35% for class 1 as compared to 60.83% and 66.73% of pre-trained BERT and Word2vec based features. The use of different machine learning (XGBoost and Random Forest), deep learning (Bi-LSTM) algorithms, and clustering-based features (K-means and spectral clustering) on top of SkillBERT is not making a statistically significant difference and the results are very similar. The difference between the validation dataset and test dataset F1 scores was less than 0.65 and 0.5 percentage points and the variance of validation data F1 scores for different hyperparameter trials was 1.20 and 1.05 percentage points for XGBoost+SkillBERT+spectral clustering and Bi-LSTM+SkillBERT+spectral clustering models respectively. We computed feature importance using the XGBoost model and "bert-prob" explained in section 2.2.1 created using SkillBERT was the top feature in the list. TFIDF and similarity-based features were also highly predictive. Next, the results of experiment 4 (core vs fringe skill classification) given in Table 5 show that we were able to classify fringe skills for a group more accurately compared to core skills. All the reported results are statistically significant at $p < 0.05$.

## 5   Conclusion

In this paper, we have addressed the problem of recruiters manually going through thousands of applications to find a suitable applicant for the posted job. To reduce the manual intervention, a competency group classification model is developed which can classify skills into multiple competency groups and hence, helps hiring managers in the quick mapping of relevant applications to a job. The difference in time is because our service which uses the SkillBERT not only matches the skills

to their competency groups, but it also shows constituent skills in each competency group. Hence the search time for skills unknown to hiring managers is reduced as they can refer to competency groups which are generic and are already known to them. Also, showing the competency group and its constituent skills helps the hiring manager in becoming aware of the competency groups to which these unknown skills belong to. However, there can still be some skills which may not be part of SkillBERT, and hence, some manual intervention may be required. Also, as our work finds the match only based on the skills mentioned by the candidates, hiring manager will still need to go through the required interview process to judge the fitment of the candidate. In the experiments, for skill representation, different word embedding models like Word2vec and BERT are used and comparisons among classification results of different machine learning models are shown. Additionally, features like TFIDF, clustering labels, and similarity-based features are explored for better classification of skills. We trained BERT on a domain-specific dataset and a significant improvement is noticed while comparing the results with pre-trained BERT and Word2vec.

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

# A   Appendix

## A.1   Spectral clustering

Spectral clustering is a widely used unsupervised learning method for clustering. In spectral clustering, the data points are treated as nodes of a graph and these nodes are then mapped to a low-dimensional space using eigenvectors of graph laplacian that can be easily segregated to form clusters. Spectral clustering utilizes three matrices, details of those are given below.

**1. Similarity graph (Affinity matrix):** A similarity graph is a pair G = (V, A), where V=$\{v_1,....,v_m\}$ is a set of nodes or vertices. Different skills are forming different nodes as shown in Figure 7. A is a symmetric matrix called the affinity matrix, such that $ba_{ij} \geq 0$ for all i,j $\in$ {1,.......,m}, and $ba_{ii} = 0$ for i = 1,.....,m. We say that a set$\{v_i, v_j\}$ is an edge if $ba_{ij} > 0$. Where $ba_{ij}$ is bert affinity between nodes i and j computed using cosine similarity between SkillBERT embeddings of the corresponding skills. The corresponding (undirected) graph (V,E) with E = $\{\{v_i, v_j\} \mid ba_{ij} > 0\}$, is called the underlying graph of G. An example of similarity graph structure as affinity matrix is shown in Figure 7.

**2. Degree matrix(D):** If A is an m×m symmetric matrix with zero diagonal entries and with the other entries $ba_{ij} \in$ R arbitrary, for any node $v_i \in$ V, the degree of $v_i$ is defined as

$$d = d(v_i) = \sum_{j=1}^{m} |ba_{ij}| \tag{1}$$

and degree matrix D as

$$D = diag(d(v_1), .........., d(v_m)) \tag{2}$$

**3. Graph laplacian (L):** If D is a diagonal matrix and A is affinity matrix then we can compute L as follows :-

$$L = D - A \tag{3}$$

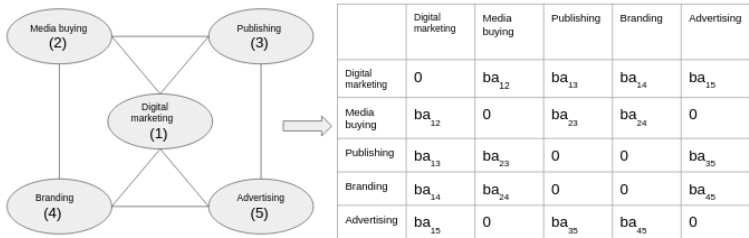

Figure 7: Adjacency matrix representation of Graph

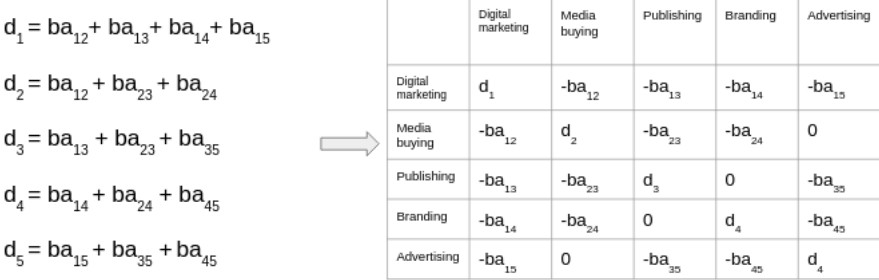

Figure 8: Graph laplacian for example in Figure 5

The Laplacian's diagonal is the degree of our nodes, and the off-diagonal is the negative edge weights (similarity between nodes). For clustering the data in more than two clusters, we have to modify our laplacian to normalize it.

$$L_{norm} = D^{-1/2}LD^{-1/2} \tag{4}$$

We know that

$$L_{norm}X = \lambda X \tag{5}$$

Where X is the eigenvector of $L_{norm}$ corresponding to eigenvalue $\lambda$. Graph Laplacian is a semi-positive definite matrix and therefore, all its eigenvalues are greater than or equals to 0. Thus, we get eigenvalues $\{\lambda_1 , \lambda_2 , ... , \lambda_n\}$ where $0 = \lambda_1 \geq \lambda_2 \geq ... \geq \lambda_n$ and eigenvectors $X_1, X_2,...,X_n$. An example of a sample laplacian matrix is given in Figure 8. Once we calculate the eigenvalues of $L_{norm}$ and eigenvectors corresponding to the smallest k eigenvalues where k is the number of clusters, we create a matrix of these eigenvectors stacking them vertically so that every node is represented by the corresponding row of this matrix and use K-means clustering to cluster these new node representations into k clusters. For our experiment, we chose the first 35 eigenvectors to create 35 clusters and used them as features for model training. The number 35 was decided using the criteria of difference between two consecutive eigenvalues. As shown in Figure 10, the difference between eigenvalue 35 and 36 is significantly bigger.

## A.2 Miscellaneous

This section contains the results of experiments done for hyperparameter selection and some figures referenced in the main text. Figure 9 shows the elbow method graph for deciding the number of clusters in K-means. Figure 10 shows the scatter plot of eigenvalues to determine number of eigenvectors and clusters in spectral clustering. Table 6 shows the results of experiments done for a varied number of top skills for similarity based features. Table 7 shows the effect of different SkillBERT embedding sizes on the results of the XGBoost classifier.

## A.3 SkillBERT training

The dataset used for training the SkillBERT model can be downloaded from here. It contains the list of skills present in job requisitions. We leveraged Bert-Base architecture on the job-skill data to

Table 6: Result for different Number of top skills similarity values in feature set (In this experiment, all the features mentioned in the experiment section "SkillBERT vs Word2vec vs Pre-trained BERT" were used and only the number of skills used for similarity value calculation were varied. As a classifier we used XGBoost)

| No. of skills used | Precision | | Recall | | F1-score | |
|---|---|---|---|---|---|---|
| | Class 0 | Class 1 | Class 0 | Class 1 | Class 0 | Class 1 |
| Top 1 skill | 99.22% | 95.15% | 98.89% | 83.92% | 99.05% | 89.18% |
| Top 2 skills | 99.27% | 96.10% | 99.26% | 84.10% | 99.26% | 89.70% |
| Top 3 skills | 99.32% | 96.65% | 99.47% | 84.82% | 99.39% | 90.35% |
| Top 4 skills | 99.21% | 95.56% | 99.40% | 84.69% | 99.30% | 89.80% |

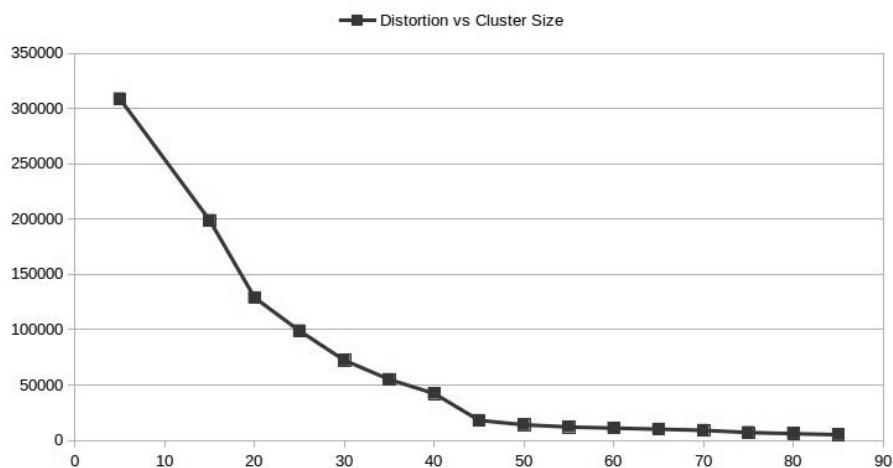

Figure 9: Elbow method graph to determine the number of clusters in K-means clustering

Table 7: Result for different embedding size (In this experiment, XGBoost was used as a classifier and bert-prob was used along with emdeddings of different sizes as independent variable. No other feature apart from these was used)

| SkillBERT embedding size | Precision | | Recall | | F1-score | |
|---|---|---|---|---|---|---|
| | Class 0 | Class 1 | Class 0 | Class 1 | Class 0 | Class 1 |
| 32 | 98.12% | 91.65% | 95.47% | 80.12% | 96.78% | 85.50% |
| 64 | 98.32% | 91.80% | 97.26% | 81.10% | 97.79% | 86.12% |
| 128 | 99.12% | 92.65% | 97.47% | 83.80% | 98.29% | 88.00% |
| 256 | 99.12% | 92.56% | 97.40% | 83.79% | 98.25% | 87.96% |

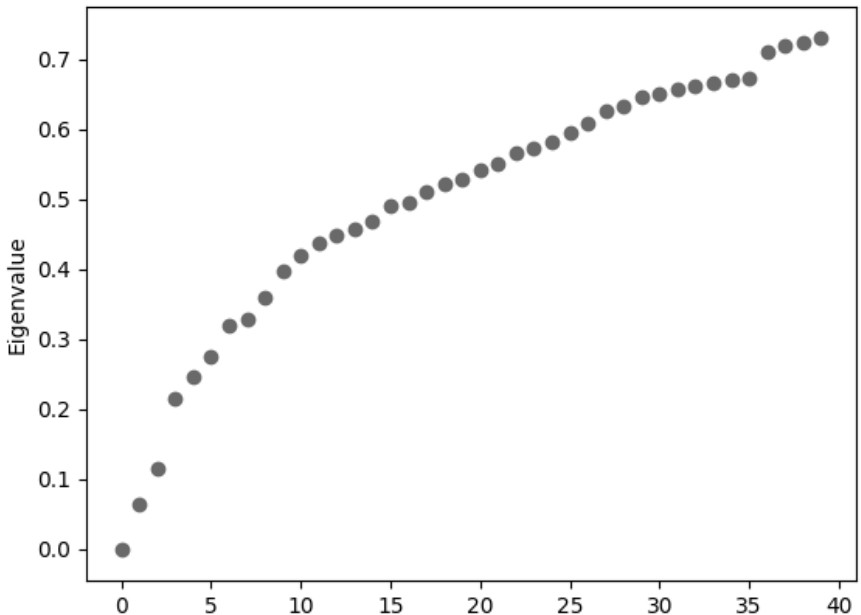

Figure 10: Scatter plot of eigenvalues to determine number of eigenvectors and clusters in spectral clustering

Table 8: Feature Description

| Feature Name | Feature Type | Dimensionality |
|---|---|---|
| bert_0 - bert_127 | SkillBERT Embedding | 128 |
| bert-prob | SkillBERT Embedding | 1 |
| 0-34 | Spectral clustering label | 35 |
| value1-value3 | skill-skill similarity | 3 |
| tf-idf | TFIDF | 1 |
| bert_grp_sim | skill-group similarity | 1 |
| core_skill_count,fringe_skill _count | group based feature | 2 |

generate embeddings of size 768, details of it can be found here. Finally, the embeddings generated using the SkillBERT model can be downloaded here.

## A.4 Features

The details of features used in the training of Bi-LSTM model, which gave us the best performance are given in Table 8 .

## A.5 Running the experiment

The code to run all the experiments mentioned in the paper can be downloaded here. This codebase uses python 3.6 and all the packages used for this experiment can be downloaded by installing requirements.txt. An overview of all the folders present in the code is given below:

**1. training_codes:** This folder contains the main python files used for running the experiments mentioned in the paper. Inside the main() method there are functions for data preparation, training, and testing. We have provided comments in each section for a better understating of the modules. The code present in the file "skillbert_spectral_clustering.py" is used to train the Bi-LSTM model on SkillBERT and spectral clustering related features which gave us the best performance. You can directly jump to this code if you don't want to run other intermediary experiments. The experiment for classifying a skill into core and fringe can be run using 3_class_classifier.py.

Apart from these if you want to run other experiments mentioned in the paper, you can do so by running "word2vec_only.py" for classifying skills using only Word2vec model, "skillbert.py" for classifying skills using only SkillBERT model, "bert_pretrain_only.py" for classifying skills using only pre-trained BERT model and "skillbert_and_kmeans.py" for classifying skills using SkillBERT and k-means on SkillBERT embedding.

**2. feature_creation:** This folder contains the code for creating features used for training the models. If you don't want to go through each code, features created using these code files are already available in the feature_data folder. Codes present in the training_code also uses these CSV files directly for the model training.

**3. feature_data:** As mentioned before, this folder contains CSV files of features generated using codes present in feature_creation folder.

**4. model:** This folder contains the final model trained using all the experiments mentioned in the paper. Folder "skill_bert_spectral_clustering" contains the Bi-LSTM model which has been used as the final model.

**5. dataset:** This folder contains the final training and testing data used for each experiment. You can use these files to directly test the corresponding model.

