# OpenReview forum: "SkillBERT: "Skilling" the BERT to classify skills using Electronic Recruitment Records"
_NeurIPS.cc/2021/Track/Datasets_and_Benchmarks/Round1 — Submitted to NeurIPS 2021 Datasets and Benchmarks Track (Round 1)_

### Official Review · Reviewer_fkNu · 2021-06-21
**Not really a benchmark paper.**

**Rating:** 2
**Confidence:** 4

**Strengths:**

 The authors provide:
- a large variety of baselines and comparisons
- explored a large number of hyper-parameters
- visualized and compared the features from BERT and Skill-BERT

**Weaknesses:**

The major weakness of this paper is the lack of detailed analysis of the collected and proposed benchmark. To me, this looks more like a main conference submission rather than the benchmark track.

Please check Correctness for more details.

**Additional Feedback:**

Nan

**Clarity:**

Some part is a bit miss-leading, but overall, it is well written.
However, the paper is 10 pages, while the limit is 9.

**Correctness:**

Several problems with the proposed dataset:
- it is very unclear where the dataset has been collected. Using words like "our organization’s database" (line 108) is very un-scientific.
- lack of detailed statistic about the dataset (e.g., how many samples per class, how long it's the description etc.)
- it is unclear who the annotators are! the authors defined them as "domain experts", but they do not specify how they trained them, and how their work has been reviewed.

**Documentation:**

Yes, the authors released their code.

**Ethics:**

Yes. The authors explicitly mentioned "the skill dataset is taken from our organization’s database", without mentioning where this data is from. It is not clear to me if this data is from customers or from individual applying to the company. If so, there is no sign of discussion, or the actual form, about whether the individual consented to the data usage.  If instead these data have been collected by crowd-sourcing, it is not clear how much the annotator has been paid and how they have been recruited.




**Relation To Prior Work:**

In the introduction, the authors clearly discuss related works.

**Summary And Contributions:**

In this paper, the authors proposed SkillBERT a BERT-based model trained on the job-skill data. These data are collected from "their" (check later weakness and questions) organization’s database, which contains 700,000 job requisitions and 2,997 unique skills, which are further clustered into 40 competency groups. The task is to classify new skills into the corresponding competency groups (multiclass classification problem). The authors compare multiple classifiers, including BERT, W2V, LSTM and Skill-BERT, and classical ML algorithms (RF, XG-Boost, etc.), and they used k-means over the Skill-BERT features. The results show that Skill-BERT features perform the best among other predictors.

---

### Official Review · Reviewer_gLy2 · 2021-07-02
**Expert-annotated job requisition dataset of little interest to the research community.**

**Rating:** 2
**Confidence:** 3

**Strengths:**

The presented dataset contains clean lists of professional skills that are either required/desired for a certain job or offered by a participant, and skills are manually assigned to more coarse competency groups. This classification might provide interesting insights for recruiters looking to automise a part of the applicant matching process.

The dataset is fully anonymised as neither jobs nor applicants are identifiable.
One of the tested models is a BERT Base model trained on only the skills dataset. Deep contextualised models like BERT are currently being investigated by a wide range of studies in order to learn more about how these models encode natural language and why they seem to perform well on different NLP benchmarks.

The parameters and resources used for the dataset-specific training of the BERT Base model as well as the application details of the different classification and clustering methods are clearly stated.


**Weaknesses:**

As the data contains only lists of skills, there is no need for language processing proper. The authors state that traditional n-gram models “completely ignore the context surrounding a word” - which does not hold true for any n>1, and I suspect that a simple unigram model would fare well on the task of assigning a skill to a competency cluster simply by using a bag-of-words representation of other listed skills. Given that this would be a naive baseline for this dataset, I recommend including this kind of simple n-gram baselines to put in perspective the high performance scores of the models tested in the paper.

Tying in to the previous point, both, the Word2Vec and the SkillBERT model are trained solely on the lists of skills – which means that not their embeddings of language use, but simply the encoding of which other skills are collocated in a list is utilised for the classification of a skill. As is, I would argue that this doesn’t provide an academically interesting challenge to the language models.

Regarding the presented classification task, the authors abstract from the task of matching an applicant to a job offering based on the required skills by generalising skills into competency groups. This could lead to higher matches between applicants and job opportunities, but might not necessarily improve the quality of the matches. Whether or not there is anything to be gained from introducing competency clusters cannot be validated with the dataset presented. There also is no guarantee of completeness in terms of the coverage of skills.

As for the training and evaluation of the models, I cannot guarantee that a skill was completely “unseen” during training, as the skills dataset lists a number of skills associated with a given applicant or job offering. During the training of the language models it might not yet have been determined which skills will be used in the test set of the competency dataset used for the classification task later, and therefore embeddings for these skills could have been generated. This suspicion seems to be supported by the statement that “a total of more than 700,000 requisitions were used for model training” - which would include the entire skill dataset.

I’m pretty sure that the ‘clusters’ in the tSNE plots of Figures 5 and 6 are hand-drawn. Please either indicate clustering by colouring or the actual decision boundaries of the classification/clustering algorithms.
Most of the engineering effort in this paper resides in the selection and application of different classification and clustering algorithms. While relevant for presenting the SkillBERT model as a product, the way it is presented in this paper is of little interest to the research community.

All in all, I’d argue that the dataset itself holds little academic interest for the language processing community, and that the presented classification task doesn’t present much significance as a benchmark for NLP systems. This is - in my opinion – further supported by the already high performance scores of the vanilla models applied to the classification task.

The paper exceeds the nine page limit.


**Additional Feedback:**

The authors mention the concept of core and fringe skills in a competency cluster. I couldn’t determine this distinction in the dataset provided.


**Clarity:**

The paper is well written and mostly clear. At some point the authors introduce the notions of 'class 0, class 1 and class 2' which are likely tied to core and fringe skills, but I couldn't reach a definitive conclusion how.


**Correctness:**

As mentioned earlier, I cannot guarantee that a skill was completely “unseen” during training, as the skills dataset lists a number of skills associated with a given applicant or job offering. During the training of the language models it might not yet have been determined which skills will be used in the test set of the competency dataset used for the classification task later, and therefore embeddings for these skills could have been generated. This suspicion seems to be supported by the statement that “a total of more than 700,000 requisitions were used for model training” - which would include the entire skill dataset. This could also have been intentional and ‘unseen’ simply means unseen during the training of the classifier – which would further reduce the relevance of the language models for the task.

The data used for training and testing the classifiers is recorded in the supplementary material – which is especially important for multi-label classification tasks like the one presented in this paper.

**Documentation:**

The dataset was generated based on data extracted from a job requisition system operated by ‘Peoplestrong’. The data has a clear structure and is easily accessible in well-established formats. The dataset is fully anonymised as neither jobs nor applicants are identifiable.


**Ethics:**

The dataset is fully anonymised as neither jobs nor applicants are identifiable.


**Relation To Prior Work:**

As the authors abstract from the task of matching applicants and job offers by introducing competency clusters, there is no directly related work. The authors do refer to previous work in the field of automatically matching entries – especially in the field of job requisitions - based on the properties of the entries (either listed or text-based).


**Summary And Contributions:**

The submission presents a dataset of 700.000 job requisitions listing a total of over 3.000 unique skills. Each skill is manually assigned to at least one of 40 so-called competency groups by domain experts. The authors then test a selection of language models (Word2Vec, pre-trained BERT Base and BERT Base trained on the skills dataset) in combination with a range of classification algorithms to predict the competency groups for an unseen skill. All tested model and classifier combinations perform remarkably well with multi-class prediction F1 scores ranging from .965 to .995.

---

### Official Review · Reviewer_NA5T · 2021-07-04
**Needs revision in writing**

**Rating:** 2
**Confidence:** 4

**Strengths:**

Human resource management is an understudied area and resources on this could be useful for NLP applications.

**Weaknesses:**

One of the main claims of the paper is to show that SkillBert is better than Bert-base. This is fairly known in the literature that domain-adaptation using fine-tuning is often better than the pre-trained versions. In fact, the “Don’t Stop Pretraining” paper popularized how pre-training (on domain-specific data) in addition to task-specific fine-tuning is useful for downstream tasks.

While the paper presents a mapped dataset from skills to competency groups, that in itself doesn’t serve as a standalone contribution to this submission track.

Many issues on the understandability of the paper and the details on the feature designs need to be better written.


**Additional Feedback:**

The introduction enlists some of the previous works related to human talent matching. However, there is no structured discussion and it is difficult to observe how the proposed work connects to this literature.

Would suggest the authors to improve the figures.


**Clarity:**

Unfortunately, the clarity of the paper is not good and needs much improvement. There are many doubts and questions that have raised in me after reading the sections.

For example, it is not clear if a single skill word is sent into BERT to get the embeddings.

While word2vec makes sense as it considers bad of words as context, using BERT (which is order-sensitive) for a bag-of-skills is not making sense to me. If alternatively, only skill word is sent into BERT, then I do not see the point of using a contextual model for it.


**Correctness:**

Presently, I am unable to see the need for using a sentence encoder, such as BERT, to encode a skill word. If the full sentence from a requisition is sent to BERT, then sending a set of works (skills) into an order-sensitive BERT seems wrong.

**Documentation:**

Code and data link is provided. However, there doesn’t seem to be any specific documentation on the data itself. Only instructions to run the code is provided (in the link and in appendix A5).

**Ethics:**

No concerns.

**Relation To Prior Work:**

I am unaware of the literature in this particular field. But the papers mentioned in the introduction should be discussed in a better structure.

**Summary And Contributions:**

The paper presents a study on the area of human resource management, which is an understudied area despite the large applications of NLP.

They propose to get embeddings of skills and groups using popular word embedding methods and further utilize other features, such as clustering, to analyze the skills.

---

### Decision · Program_Chairs · 2021-07-26

**Decision:**

Reject

**Comment:**

There is strong agreement amongst reviewers that this paper is not a good fit  for the track. Reviewers identify some issues relating to clarity and misleading text. The primary concern however is the relevance to the track, with reviewers noting the paper’s focus lies largely in the comparison of algorithmic methods. Moreover, many details of dataset development are omitted making it ill-suited for a data-set focused track.

On the positive side, reviewers note the extensive empirical evaluations as a strength of the paper.